# Improve Model Testing by Integrating Bounded Model Checking and Coverage Guided Fuzzing

Yixiao Yang 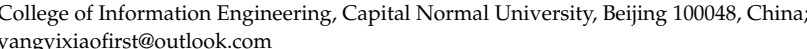

College of Information Engineering, Capital Normal University, Beijing 100048, China;
yangyixiaofirst@outlook.com

**Abstract:** Eectromechanical systems built by Simulink or Ptolemy have been widely used in industry fields, such as autonomous systems and robotics. It is an urgent need to ensure the safety and security of those systems. Test case generation technologies are widely used to ensure the safety and security. State-of-the-art testing tools employ model-checking techniques or search-based methods to generate test cases. Traditional search-based techniques based on Simulink simulation are plagued by problems such as low speed and high overhead. Traditional model-checking techniques such as symbolic execution have limited performance when dealing with nonlinear elements and complex loops. Recently, coverage guided fuzzing technologies are known to be effective for test case generation, due to their high efficiency and impressive effects over complex branches of loops. In this paper, we apply fuzzing methods to improve model testing and demonstrate the effectiveness. The fuzzing methods aim to cover more program branches by mutating valuable seeds. Inspired by this feature, we propose a novel integration technology SPsCGF, which leverages bounded model checking for symbolic execution to generate test cases as initial seeds and then conduct fuzzing based upon these worthy seeds. Over the evaluated benchmarks which consist of industrial cases, SPsCGF could achieve 8% to 38% higher model coverage and 3x-10x time efficiency compared with the state-of-the-art works.

**Keywords:** model testing; testing of electromechanical systems; testing of advanced autonomous systems

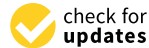



## 1. Introduction

MATLAB Simulink [1] is the most widely used modeling tool, which supports physical simulation, automatic code generation, formal verification, and continuous testing for complex control systems, such as robotics and autonomous systems. It has also been widely deployed in a number of domains, such as biotechnology, chemical industry, finance, and so on. Recently, there have been many research hotspots in these fields, such as research which analyzes computational efficiency of multi-body systems [2] and personnel task allocation [3] based on intelligent decision making. After modeling the physical world by Simulink, engineers need to check the model to find possible safety or security problems. The automatic test generation and verification technologies based on Simulink have been proposed to help engineers to achieve their goals. Although existing state-of-the-art methods achieved considerable performance in small or medium scenarios, they still face certain challenges when deployed to industrial cases.

One challenge is that when handling large scale complex industrial cases, the existing methods may fail to generate test cases, which can lead to low coverage. Simulink models are highly complex; they may contain a large number of blocks and hierarchy levels. In large industrial cases, the resulting Simulink models often consist of more than ten thousand blocks and hundreds of hierarchy levels. This exposes hard problems for model testing and validation. The execution of the Simulink model is not only time driven, but also event driven [4]. The time-event mixed computing makes the testing and validation even harder. For model verification methods, they suffer from the state space explosion problem,

making them nearly impossible to be applied to complex industrial models. Among tools based on verification methods [5–10], the most grinding one is Simulink Design Verifier [10]. Simulink Design Verifier converts the Simulink model into a linear hybrid system [11] and uses formal methods to analyze that system. The test case generation functionality of Simulink Design Verifier cannot handle nonlinear elements. If a cycle exists in a Stateflow chart or an integrator exists in the Simulink model, Simulink Design Verifier will fail and the semi-finished coverage report will be given. However, for linear elements, the Simulink Design Verifier can achieve good results. Recently, search-based model-testing methods have become popular to find safety and security matters. They are light-weight and effective, capable of discovering various kinds of bugs when combined with bug-detection algorithms.

Even with the introduction of search based methods, existing methods still face two problems: slow testing speed and low testing efficiency. Existing available tools are based on Simulink simulation, which requires additional overhead and does not support parallel execution. These tools have serious efficiency problems when applied to large industrial cases. For search-based model testing tools [12–19], they basically define fitness values according to the testing objective and leverage genetic algorithms to search for test cases with the highest fitness value. Recently, coverage-guided fuzzing methods have been proposed not only to achieve high coverage but also to test the model in pressure to discover possible defects. The fuzzing methods are extremely fast. Tens of thousands of inputs can be generated and executed in a second. In this paper, we try to apply the coverage guided fuzzing methods to model testing and combine the traditional bounded model checking methods together. The fuzzing method was originally used for source-code testing. To use that method in model testing, the model is transferred to the source code first and the fuzzing method is applied to the code of the model. However, employing fuzzing techniques for model testing still requires adaptations. Fuzzing techniques randomly choose predefined mutation operators to generate the signal value at each time step. These techniques often lead to irregular signal waves. The irregular signal waves make the final coverage unsatisfactory. Thus, traditional source-code testing techniques need to be improved. To improve the effectiveness of the fuzzing method, based on original mutation operators of fuzzing methods, we summarize the commonly used signal patterns and design mutation operators to mutate signal data.

In this paper, to improve the testing speed and the testing efficiency, we put forward a novel model testing framework SPsCGF, which is composed of two stages. The first stage is to use bounded model checking to collect and solve all path constraints in the control logic to generate initial test cases. The loop in the control logic is expanded a finite number of times. The second stage is to use coverage guided fuzzing techniques to mutate the initial test cases in the first stage to cover the uncovered branches. The guided fuzzing methods not only explore the program paths as many as possible but also explore the extreme boundary conditions of the program to ensure the safety and security. Through the two stages, we combine the advantages of existing bounded model-checking methods and fuzzing methods together. The experiments prove the effectiveness of the proposed framework. Compared with state-of-the-art tools, the proposed method performs 3×–10× faster. The main reason for the efficiency improvement is that existing search-based tools are based on Simulink simulation while the proposed framework is based on raw code execution. The Simulink simulation requires additional overhead compared with raw code execution, and does not support parallel execution. According to experiments, the proposed framework can achieve 8% to 38% coverage improvement compared to state-of-the-art tools.

In summary, the contributions are as follows:

(1) We developed SPsCGF, an integration method which integrates the bounded model-checking method and coverage guided fuzzing method together to generate test cases for models with signal inputs. In coverage guided fuzzing modules, SPsCGF takes signal patterns into consideration to improve the mutation operators of fuzzing meth-

ods. We have implemented SPsCGF as a standalone application. The SPsCGF relies on the existing state-of-the-art model checking tool CBMC [20] as well as AFL [21], a state-of-the-art, open-source coverage guided fuzzing framework.

(2) We compared SPsCGF with two state-of-the-art baselines to assess the effectiveness and efficiency of the proposed testing method. Our experiments, performed on 12 publicly available Simulink models from the commonly used benchmarks and industry cases, show that, on average, SPsCGF achieves 8% to 38% more coverage and performs $3\times$ to $10\times$ faster than state-of-art baselines.

(3) We evaluated the usefulness of SPsCGF in real large industrial CPS models from the vehicle control domain. Those models are real industry models from our industrial partner. Those models have different formats and run under different environments. SPsCGF successfully tests those models and achieves high coverage.

## 2. Model Testing with Time Series Data Input

In this section, we describe the difference between traditional source code testing and model testing. Then, we formulate the problems. We use a real case to show the difficulties about testing Simulink models with time series data inputs. This explains why we need to specially design mutation operators based on signal patterns in fuzzing.

Take a model which simulates the forced shutdown of an electronic device in automobile as an example to show why models with time series data inputs are difficult to test. The model is shown in Figure 1. This model is a simple molecular model of the original big embedded model which is actually used in industrial scene. We intercept the small core part to show its key control logic. In Figure 1, the input *u* represents the signal about button pressing down, the input *Tset* indicates how long the electronic device will be forced to shut down after the shutdown key is pressed and held. At each time the input signal arrives, the *ONDLC* module is responsible for generating the output signal. The output signal of the model will be used as the input of the model in the next round. The core logic of *ONDLC* is also given in Figure 1b. If the input signal *u* at the current time is 10, the counter will increase by 1, otherwise the counter will be reset to 0. If the accumulated value of the counter in the past time reaches the threshold specified by *Tset* port, the output signal will be true and shows that the electronic device will be forced to shut down.

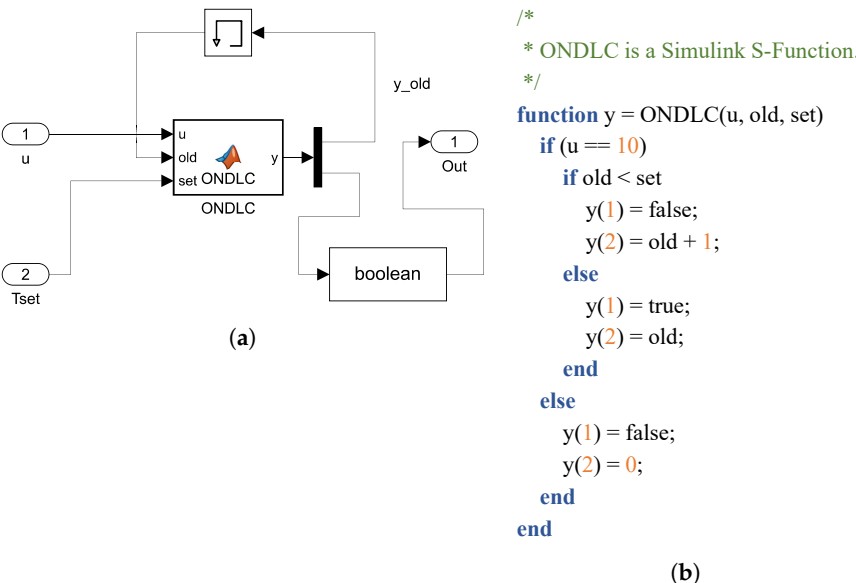

**Figure 1.** Model example with signal unput. (**a**) model; (**b**) code of ONDLC in model.

Although this logic is simple and clear, this control logic cannot be directly solved by symbolic execution as the impact of *u* to the final output is indirect. The symbolic execution can only solve the constraint $u == 10$ at each time step. If the number of total time steps is *n*,

the symbolic execution will produce $2^n$ results. However, only one kind of result in a total of $2^n$ results can make the final output to be *true*. That is, u keeps being 10 for N (specified by *Tset*) time steps and is set to 0 for other time steps. For bounded model checking, the loops are only expanded for a finite number of times. If $n$ is larger than the number of expanding times, bounded model checking cannot give useful results. This core model is only a part of the original big embedded model. Things become more complicated if we consider the original big embedded model. If the taint analysis technique in search based test generation is used, it will find that each $u$ in each time step has influences for the final results. Thus, those techniques still cannot efficiently generate test cases for models with signals as inputs. For models which receive curve signals, traditional testing techniques perform much worse. Thus, it is necessary to handle signals specially.

## 3. Overall Framework

Figure 2 shows the overall workflow chart. Given a Simulink model, as shown in Figure 2, there are four steps to test that model. In the first step, the model is translated into code. The code format is C++. To be compatible with Simulink and Ptolemy, in this step, actually, the model is translated into an intermediate representation (IR). This IR is an intermediate model representation which contains both Simulink and Ptolemy semantics. Then, we design a code generator which generates C++ code based on IR. Note that this IR can directly contain source code. For complex Simulink elements, we directly put the code of those elements into IR. In the second step, based on the C/C++ code, the initial test cases are generated. The tool uses bounded model checking for symbolic execution to generate test cases. The detailed components about initial test case generation are shown in the left part of Figure 3. The third step is to do coverage guided fuzzing. The detailed components about coverage guided fuzzing are shown in the right part of Figure 3. The last step is to measure the generated test cases and update the test case pool. The details are described in the following paragraphs.

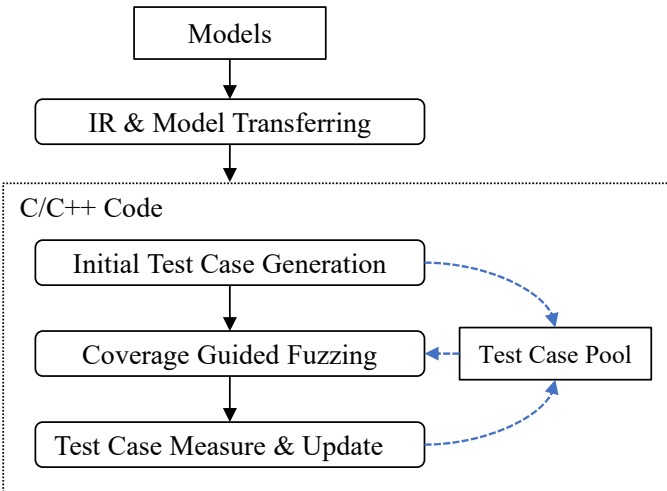

**Figure 2.** Overall Workflow Chart.

In the steps just described, the initial test case generation module and the coverage guided fuzzing module are two core modules of the proposed testing framework. As shown in Figure 3, when generating initial test cases, we not only use bounded model checking to carry out symbolic execution, but also use the n-wise algorithm to generate test inputs. The n-wise algorithm ensures that all combinations of all possible values of n ports must exist in the test cases. This algorithm is expensive, so it is only applied on constant ports. After generating initial test cases, we put the generated test cases into a test case pool. The coverage guided fuzzing selects a test case from the test case pool at each time and mutates that test case for multiple rounds to generate new test cases. The

remaining subsections correspond to the four steps in the workflow chart in Figure 2 one by one. The technical details including the algorithms and the important implementation tricks are described in the following subsections.

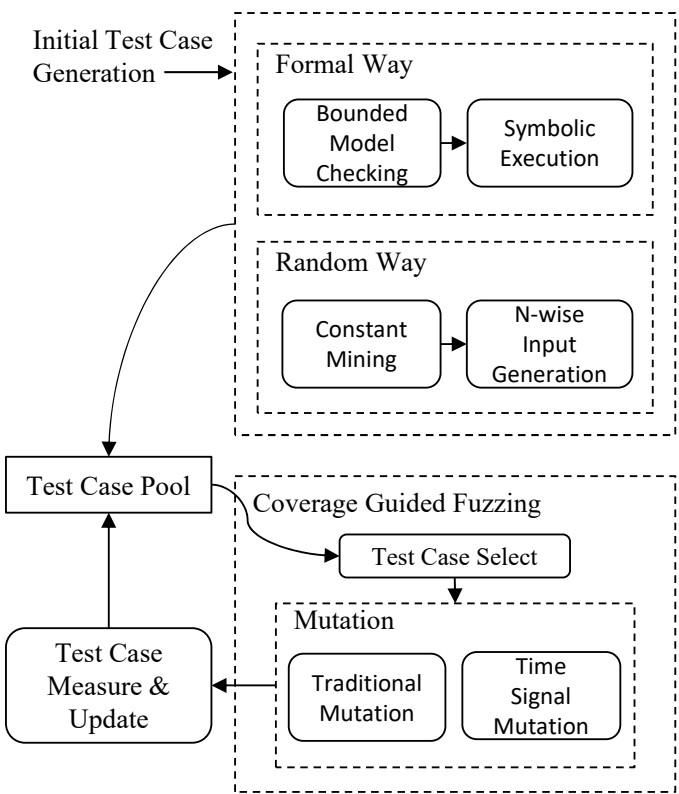

**Figure 3.** Components of modules in workflow chart.

### 3.1. IR and Model Transferring

To be compatible with Simulink and Ptolemy, we design an intermediate representation (IR) of the model. This IR can be considered as a simple modification of the Simulink format and can be compatible with the specific features of Ptolemy. Based on IR, we implement a code generator which generates runnable code. The code generator is also responsible for generating the test driver for bounded model checking and coverage-guided fuzzing. The IR can directly contain source code. For complex Simulink models, we cannot easily translate those models into IR, thus, we directly use Simulink Coder to generate code and include the code in IR. The libraries used by the code are also added to IR. The self implemented code generator generates code based on IR.

Figure 4 shows a simple Simulink model and its corresponding IR. Figure 5 shows the corresponding C/C++ code for the IR in Figure 4. The new work in the code generator is to generate test drivers for two main modules of the proposed framework: the bounded model checking and coverage-guided fuzzing. For bounded model checking, the test driver is responsible for assigning the symbol indicator to variables corresponding to input ports so that the bounded model checking tool CBMC can recognize the symbol indicator to carry out symbolic execution. CBMC can only handle C code, so the test driver for CBMC is in strict C format. For fuzzing, the existing fuzzing tool only accepts a byte buffer as a test case. Thus, the test case in our fuzzing module is also a byte buffer. We splice the data of all input ports into a byte buffer. The position and length of each port data in that buffer are recorded. After a test case (byte buffer) is generated, the test driver is responsible for assigning values in the byte buffer to variables corresponding to input ports. After the value assignment, functions which use those variables can run properly.

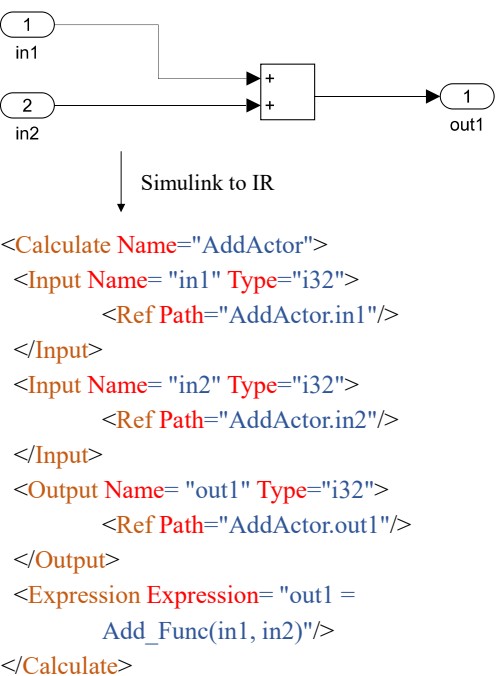

```
<Calculate Name="AddActor">
  <Input Name= "in1" Type="i32">
          <Ref Path="AddActor.in1"/>
  </Input>
  <Input Name= "in2" Type="i32">
          <Ref Path="AddActor.in2"/>
  </Input>
  <Output Name= "out1" Type="i32">
          <Ref Path="AddActor.out1"/>
  </Output>
  <Expression Expression= "out1 =
          Add_Func(in1, in2)"/>
</Calculate>
```

**Figure 4.** Simulink to IR.

Figure 5 shows how the test drivers assign the symbol indicators to the input ports. In Figure 5a, the function *nondet_int* is the CBMC utility function which is responsible for registering symbols in CBMC symbolic executor. The *nondet_int* function is responsible for marking which data is the symbol. The *__CPROVER_input* function is just to provide some additional comments and explanations for the data which is marked by *nondet_int* function. The example only shows the data in one time step for simplicity. in1 and in2 are two input ports of the model and out1 is the output port. After marking in1 and in2 as symbols, the next step is just to pass in in1 and in2 as parameters of function *AddActor* corresponding to the model logic. In Figure 5b, the *buffer* is the test case generated by fuzzing engine. How to generate this data will be described in the following section. The driver assigns the data in the buffer to each port using *memcpy* function. The names of input ports in code and the names of input ports in the model are the same in this example.

```
/* Model code and CBMC driver. */
// input data for one time step.
int in1[1];
int in2[1];
int out1[1];
// set up symbols for symbolic execution.
*((int*) in1+0) = nondet_int();
__CPROVER_input("int", *((int*) in1+0));
*((int*) in2+1) = nondet_int();
__CPROVER_input("int", *((int*) in2+1));
// target function code.
AddFunc(in1[0],in2[0],&(out1[0]));
```

(**a**)

```
/* Model code and Fuzzing driver. */
// inputs generated by fuzzing engine.
byte buffer[buffLen];
// input data for one time step.
int in1[1];
int in2[1];
int out1[1];
// assign buffer data to each port.
memcpy(in1, buffer + 0, 4);
memcpy(in2, buffer + 4, 4);
// target function code.
AddFunc(in1[0],in2[0],&(out1[0]));
```

(**b**)

**Figure 5.** IR to C/C++ Code. (**a**) IR to C code with test driver for CBMC; (**b**) IR to C++ code with test driver for fuzzing.

*3.2. Initial Test Case Generation*

Based on the driver for CBMC, we can directly use CBMC to perform symbolic execution to get the desired test cases. The CBMC tool not only generates test cases aimed for statement coverage or branch coverage, but also generates test cases aimed for modified condition/decision coverage (MC/DC) coverage. By using CBMC, a large amount of workloads can be reduced. However, only using CBMC to generate test cases is not enough. Looking back at the motivation example model mentioned in Figure 1, even for the model without nonlinear elements or external library dependencies, the symbolic execution cannot solve the branches which have indirect impacts on the input ports. If the model contains nonlinear elements, there is a high probability that the symbolic executor will not generate any test case. If the model contains arithmetic components that involve floating-point numbers, there are also probabilities that the symbolic executor cannot solve the constraint. In extreme cases, CBMC may generate only one or two test cases if the model is not suitable for bounded model checking. Thus, the fuzzing method plays an important role for the coverage improvement.

Some ports are not signals but only constant values. For those input ports or parameters, if the candidate values are given, we use n-wise algorithm to generate the combination of values of ports. In actual industrial scenario, the company will force such n-wise testing. The n-wise algorithm ensures that the value combinations of every n ports must appear in the test cases. We figure out a fast algorithm which is based on dynamic programming. Algorithm 1 uses a recursive function *FastNWise* to show the details. The symbol $\times$ in Algorithm 1 is the Cartesian product. Assume each element in a set is a list. If an element is a single value, it is taken as a list containing that single value. The Cartesian product of such two sets still produces a set which consists of lists. The *FastNWise* in Algorithm 1 accepts two parameters. The first parameter is **n** of **n**-wise. The second parameter is the index of the constant port. For constant ports, we give them index from 0 to **k**. The **k+1** is the total number of constant ports. The return value of *FastNWise*(**n**, **k**) is the desired all n-wise combinations. The return value of *FastNWise* is a set. Each element in that set is a list. In that list, each element is the value of a port. Now we describe how we cut the problem into sub-problems. The **n**-wise combination of 0 to **k** ports depends on the the **n**-wise combination of 0 to **k-1** ports. The **n**-wise combination of 0 to **k** ports also depends on the (**n-1**)-wise combination of 0 to **k-1** ports. By cutting the problem into sub-problems in this way, we can easily obtain an efficient implementation. The time complexity of the proposed algorithm is $O(n*k*v)$ while the time complexity of the naive implementation is $O((C_k^n*v)^2)$. Here $n$ and $k$ are introduced in this paragraph, $v$ is the average number of candidate values of ports. The proposed algorithm is fast. After generating initial test cases, we put those generated test cases into a test case pool which will be used in fuzzing. Since the combination of port values generated by the n-wise method will increase exponentially with the growth of n, we guarantee the combination of 2-wise (pairwise) in the actual industrial scenario.

---

**Algorithm 1** FastNWise(n, portIndex)

---

**Require:** two parameters;
  1: n is the desired n-wise;
  2: portIndex is the relative index of all constant ports;
**Ensure:** the test cases $C_n$ which satisfy n-wise;
  3: S = all constant values of port at portIndex;
  4: a = randomly select an element in S;
  5: R = S - a;
  6: $P_n$ = {a} $\times$ FastNWise(n, portIndex-1);
  7: $Q_n$ = R $\times$ FastNWise(n-1, portIndex-1);
  8: $C_n = P_n \cup Q_n$;
  9: **return** $C_n$;

---

*3.3. Coverage Guided Fuzzing*

We maintain a test case pool which was initially filled in test cases generated by previous subsection. With the test case pool, in each fuzzing turn, we select a test case which is also named as seed from the test case pool. The test case pool is also named as seed pool. After a seed is selected, then, the seed is mutated to generate new seeds. The new seeds which trigger new program paths are retained and added to the seed pool. The overall framework of the fuzzing procedure is described in Figure 6. The two important modules in fuzzing are **seed selection** and **mutation**. The mutation is to generate new seeds based on a set of given mutation operators. The mutation operators are named as *mutators* in this paper. A *mutator* can change a seed to varying degrees. The pool which contains the predefined mutation operators is named as *mutator pool*. In rest of this section, the **seed selection** and **mutation** will be described in detail.

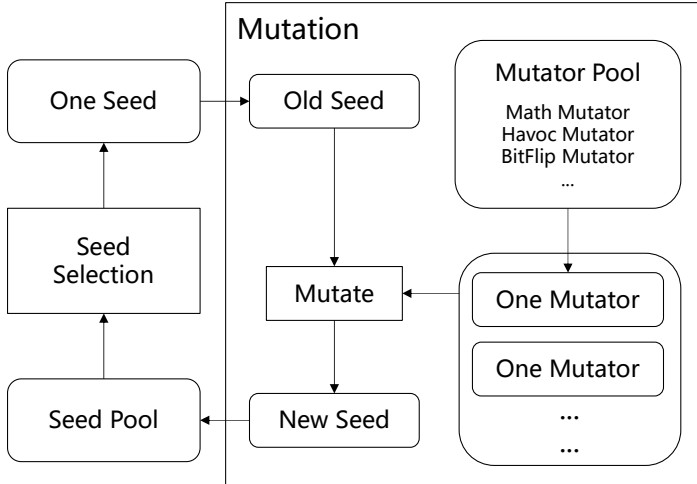

**Figure 6.** Overall fuzzing work flow.

3.3.1. Seed Selection

The seed-selection algorithm takes the selection times of a seed and the execution times of half flipped conditions in a seed into consideration. The details are described in Algorithm 2. The function *SelectTimes* in Algorithm 2 returns the number of times the test case is selected for fuzzing. The function *ExecTimes* is responsible for returning the execution times of a half flipped condition in all test case executions. If the True state of a condition is covered while the False state is not covered, this condition is the half-flipped condition. If the False state of a condition is covered while the True state is not covered, this condition is also the half-flipped condition. Existing methods often guide testing using the distance to uncovered branches, this guidance is not enough because in many cases, a branch (also named as decision) contains two or more conditions. Only considering branches will lose information of conditions in a branch. Thus, we use conditions to guide the testing. The basic idea of selecting a seed in Algorithm 2 is that if a seed is selected fewer times, it has a higher probability of being selected. If the half-flipped condition in a seed is executed fewer times, that seed also has a higher probability of being selected.

After selecting a seed, that seed will be mutated multiple times. The mutations fall into two categories: traditional mutation and time series data mutation. The traditional mutation inherits from mutation operators in AFL. The time series data mutation is designed to generate square or curve shaped input signal. The details are defined as follows.

---

**Algorithm 2** SeedSelect(pool)

---

**Require:** one parameter;
 1: pool is the test case pool;
**Ensure:** one suitable test case;
 2: distribute = [];
 3: **for** each case $\in$ pool **do**
 4:     prob = $\frac{1}{1+SelectTimes(case)}$;
 5:     prob2 = 0;
 6:     counter = 0;
 7:     **for** *condition* $\in$ half flipped conditions **do**
 8:         prob2 += $\frac{1}{1+ExecTimes(condition)}$;
 9:         counter++;
 10:     **end for**
 11:     case_prob = prob * (prob2 / counter);
 12:     distribute.append(case_prob);
 13: **end for**
 14: *index* = Sample(distribute);
 15: **return** *index*$_{th}$ test case in pool;

---

3.3.2. Mutation

As shown in Figure 6, a seed can be changed by a *mutator*. The *mutator* is a mutation operator which changes the seed. The *mutator pool* contains all predefined mutation operators. There are a large number of *mutators* that we will introduce one by one. Before the mutation is introduced, the seed to be mutated will be described first.

**Seed.** The seed to be mutated can have different formats according to different testing scenarios. The seed can be a raw primitive data array, a PDF file, a protocol packet or a text file. To ease the description of the mutation, the seed format can be uniformly described as a byte array regardless of the original seed format.

$$seed\_format : [byte, byte, byte, byte, byte, byte, ..., byte]$$

**Traditional Mutators.** The traditional *mutators* contain *RandomSet mutator*, *BitFlip mutator*, *Math mutator* and *Havoc mutator*. The details of are described as follows.

(1) **RandomSet Mutator.** This *mutator* randomly chooses a number of positions and changes the values on those positions to another random values. The random values can be the mined constant values mentioned in previous subsections. The value ranges of ports are taken into consideration. When generating random values for those ports, if some positions in test input have the constraint of value ranges, we will randomly set values according to ranges. For example, for a model with 4 ports, each port takes an integer as input, these 4 integers are filled into a byte array names as *seed_4_ports*, the first 4 bytes in *seed_4_ports* which represents the integer value for 0th port can be randomly set to a random value in [0,10] if the value range of 0th port is [0,10].

(2) **BitFlip Mutator.** This *mutator* flips a seed by bits. The smallest unit to bit flip is a byte. We flip *n* bytes each time and *n* is a random number. For example, if we flip a byte in a seed, this changes the original seed byte from 01110001 to 10001110. The non-deterministic mutation randomly selects bytes for flipping.The deterministic mutation flips all bytes in a seed. We combine deterministic mutations and non-deterministic mutations together.

(3) **Math Mutator.** This *mutator* randomly selects a number of positions in a seed to do mathematical operations. The operations contain commonly used math operations such as add, subtract, multiply, and divide. The operands are selected from 1 to a specified MAX value. For example, if a seed *seed_arr* is a byte array of length 4, if the math *mutator* is applied to the 2*th* byte of *seed_arr*, then *seed_arr*[2] += 14. The value 14 is also randomly chosen for mutation.

|     |                                          |
| --- | ---------------------------------------- |
| +   | +1, +2, +3, +4 ..., + MAX                |
| −   | −1, −2, −3, −4, ..., −MAX                 |
| *   | *2, *3, *4, ..., *MAX                     |
| /   | /2, /3, /4, ..., /MAX                     |

(4) **Havoc Mutator.** This *mutator* is for destroying a seed as much as possible. The destruction includes setting some data in a seed to specific extreme values such as −128 and 127 for a char type value, −32,768 and 32,767 for a short type value and so on. If the data type of a seed cannot be determined, some consecutive bytes of the seed are randomly selected, and all bits in them are assigned 1 or 0. The destruction also includes swapping parts of a single seed and crossing over two seeds. To cross over two seeds, we divide each seed into the same two parts, and then randomly splice these divided parts. For example, if seed *a* is [1,5,6], seed *b* is [127,−12,6,3], the *a* is split into [1,5] and [6], the *b* is split into [127,−12] and [6,3], the cross over result can be [1,5,6,3] and [127,−12, 6].

**Time Series Data Mutator.** The *mutators* for time series data divide into two categories: square signal *mutator* and curve signal *mutator*. The first one is responsible for generating histogram shape signals. The second one is responsible for generating curve shape signals. The details are as follows.

(1) **Square Signal Mutator.** The square signal *mutator* does not strictly force the signal to be the shape of a square. Instead, this mutator randomly chooses a continuous piece of data in test input and makes all the chosen data to the same value. This *mutator* is applied when the inputs of some ports are time series data. For example, if the original seed is [1,5,127,−28], then, the mutated seed can be [1,10,10,−28], the middle piece of data is set to a fixed random value.

(2) **Curve Signal Mutator.** The curve signal *mutator* randomly chooses a continuous piece of data and sets the data to the randomly generated continuous curve. The curve generating algorithm is as follows: $result = \sum_{i=0}^{n_1} sin(rand() * x + rand()) + \sum_{j=0}^{n_2} cos(rand() * x + rand())$. The generated curve will be randomly truncated to be the same length as the data chosen for mutation. For example, if the original seed is [1,5,127,−28], then, the mutated seed can be [1,sin(0.1),sin(0.2),−28], the middle piece of data is set to a piece of a curve signal wave *sin*.

All the above mutations are randomly chosen for one or multiple times when mutating a test case. This means that the final generated test case may be a complete square signal, a complete curve signal, or a combination of these two signals. The coverage of each newly generated test case will be recorded to determine whether to discard the test case and how to update the test case pool. This will be described in the following subsection.

*3.4. Test Case Measure and Update*

In this section, we describe how we instrument the code and measure test cases. We use three kinds of code coverage to measure test cases: model unit coverage, condition/decision coverage, and modified condition/decision (MC/DC) coverage. Ptolemy and Simulink also allow users to insert custom C or C++ code into the model; the proposed method can also be applied to the general C or C++ source code. In our practice, we find that there are many difficulties in source code instrumentation. Some Boolean expressions may have side effects, for example, $a < b + +$. The execution of this expression will affect the data in memory. This will cause problems if we use the traditional source code instrumentation method: each condition in a large decision in the source code is extracted separately in advance and assigned to a Boolean variable. Then, the result of the Boolean variable will be recorded. In most cases, the traditional source code instrumentation method works fine but it will fail when the Boolean expressions have short circuit characteristics. Take the Boolean expression $c < d \;||\; a < b + +$ as an example. The $c < d$ is extracted as a variable and instrumented as $bool\ b1 = c < d$, $record(b1)$. The $a < b + +$ is extracted as a variable and instrumented as $bool\ b2 = a < b + +$, $record(b2)$. The expression $c < d \;||\; a < b + +$

is rewritten as $b1 \, || \, b2$. Before executing $b1 \, || \, b2$, two expressions *bool b1 = c < d* and *bool b2 = a < b + +* have already been executed. However, because of the short circuit characteristics of Boolean expressions, if $c < d$ is true, $a < b + +$ should not be executed. Executing $a < b + +$ will cause data inconsistency between the original code and the instrumented code. To overcome this problem, we design a novel lightweight source code instrumentation method which can be applied to general C or C++ source code to collect MC/DC coverage.

For MC/DC coverage, we use a novel technique to carry out instrumentation. To insert the instrumentation code to the boolean expression, we utilize the ternary operator. For each boolean expression *e*, we rewrite *e* into *e ? record(true, cond_index, dec_index): record(false, cond_index, dec_index)*. The *cond_index* is the condition index (described later) of expression *e*. The *dec_index* is the decision index (described later) of the decision expression (described later) containing expression *e*; A Boolean expression *e* must be either a decision or a condition. A Boolean expression *e* is a decision, meaning that the boolean expression *e* is not contained by any other boolean expressions, otherwise, a boolean expression *e* is a condition. For each decision, we assign a global index to that decision. This index is referred to as *dec_index*. If a boolean expression *e* is a decision, we give the condition index 0 to that expression. For each child condition in a decision, we assign an index (starts from 1) to that condition. This index is referred to as *cond_index*. The condition index only considers all conditions in a decision. The record function is defined in Algorithm 3. By using Algorithm 3, the boolean result of a decision and boolean results of all conditions in that decision can be recorded in a single integer and that integer is stored in a slot of *bitmap* array. Because the result of a Boolean expression is either 0 or 1, we can use bits in a long long integer to store the result of Boolean expressions in a decision. A long long integer can only store 64 conditions in a decision, if there are more than 64 conditions in a decision, we use more long long integers. Here, we only use one long long integer to show the algorithm for simplicity. Then, after executing test case *t*1, assume the final bitmap generated according to the above description is *bitmap*1. After executing test case *t*2, assume the final generated bitmap is *bitmap*2. For a decision with decision index *d_idx*, the decision is flipped (decision results are different in two test cases *t*1 and *t*2) can be computed by *dec_flipped = (((bitmap1[d_idx] xor bitmap2[d_idx]) & 0x1) == 1)*. The condition in that decision with index *cond_idx* is flipped and all other conditions are not flipped can be computed by *cond_flipped = ((bitmap1[d_idx] ≫ 1 xor bitmap2[d_idx] ≫ 1) == 1 ≪ (cond_idx - 1))*. If the decision is flipped and one condition is flipped with all other conditions not flipped, one MC/DC condition is covered.

---

**Algorithm 3** record(res, cond_index, dec_index)

---

**Require:** two parameters;
  1: res is the boolean result of expression e;
  2: cond_index is the condition index of expression e;
  3: dec_index is the decision index of the expression containing expression e;
**Ensure:** record the result: res;
  4: $bitmap[dec\_index] \, | = res << cond\_index$;
  5: **return** *res*;

---

When updating the test case pool, if a test case increases the model unit coverage or condition/decision coverage or MC/DC coverage, then, this test case is considered interesting and will be added to the test case pool. For each test case, we record each specific branch or condition it covers. We generate a signature for all branches and conditions a test case covers. When we get the signature of a new test case, we will compare the signature with the signatures of all the test cases in the test case pool. If the signature of that new test case never appears, that new test case will also be added to test case pool. Otherwise, that test case will be dropped.

## 4. Evaluation

In this section, we empirically evaluate the proposed framework by answering the following research questions:

**Effectiveness—RQ1**: Can this proposed method improve the coverage compared to baselines? How much coverage can this proposed method improve?

**Efficiency—RQ2**: Can this method improve the running speed compared to baselines? How much faster can this method run?

**Usefulness—RQ3**: In the actual large-scale model, do the newly proposed mutation operators based on signal patterns have any effect? Can the fuzzing module improve MC/DC coverage?

**Implementations.** This tool is a standalone cross-platform application which contains 26,000 lines of C++ code. Both Windows and Linux platforms are supported. Associated tools are provided, including Simulink model code to IR tool and an embedded self-written C code parser. The materials (https://github.com/EmbedSystemTest/SimulinkTest, accessed on 15 February 2023) including benchmarks and the academic version of the tool are publicly available. For parallel execution, multiple fuzzing processes are started, and they can communicate with each other by sharing the same test case pool.

**The baselines.** One of the baselines is the Simulink Design Verifier (SLDV). Because SLDV [10] is a famous testing and verification tool which supports both the static analysis and dynamic analysis. Furthermore, SLDV is a commercial tool which has a user base of tens of millions. Another baseline tool is an academic tool. There are many academic tools in the last decades but most of them are unavailable now. The recently available tool is SimCoTest [22] which uses genetic algorithms to generate test cases for Simulink models.

**The models in benchmark.** At present, all mainstream tools are based on Simulink rather than Ptolemy, so here, we only use Simulink models for comparative experiments. For non-CI-CPS models, we used the publicly available benchmarks (i.e., RHB, AT, AFC, IGC) that have been previously used in the literature on testing of CPS models [23–27]. The models represent realistic CPS systems from different domains, including IoT, smart home and automobile. As the models contain state machines and continuous behaviors, we must use Simulink Rapid Simulation Target in Simulink Coder to generate the code under testing. By configuring the solver to be in fixed step mode, this target can generate code interacting with the Simulink real time library to simulate the real time environment. In addition to non-CI-CPS models, we also include pure control logic models (ie. NLGuidance, Euler321, BasicTwoTanks, EB, Regulator) in control fields such as fuel control, road control based on Euler distance and neural network guidance. These models are previously used in the Simulink verification survey [28]. The MHI1209 model is an industry model from the Mitsubishi Heavy Industries (MHI) company. The key modules related to the company technology have been deleted by engineers in MHI and can be used academically.

### 4.1. RQ1—Effectiveness

A key challenge regarding the empirical evaluation of SPsCGF is that both SPsCGF and SimCoTest rely on randomized algorithms. Hence, we have to repeat our experiments numerous times for different models. To compare the SPsCGF with baselines, the condition and decision coverage are collected. As the testing report of SLDV takes all unique conditions or decisions as the objectives, we follow the same rule to compute the coverage. Table 1 shows the coverage values for SPsCGF-BMC, SPsCGF, SLDV and SimCoTest. The SPsCGF-BMC is part of SPsCGF. SPsCGF-BMC only uses bounded model checking to generate initial test cases, and does not contain the fuzzing module. The SPsCGF is the proposed tool which does fuzzing based on initial test cases generated by SPsCGF-BMC. All tools are configured to run for 60 s. However, even we set the rules, SLDV and SimCoTest seldom obey the rules; they often take more than 180 s to finish running.

For non-CI-CPS models, SLDV cannot generate a single test case. The coverage achieved by SLDV is 0%. The reason is that there exists an integrator in the model. The

symbolic executor in SLDV cannot handle the integrator. There are also other elements SLDV cannot solve, for example, the event dispatch in the loop of Stateflow or complex embedded C code. In the meanwhile, SPsCGF performs significantly better than SLDV. For complex non-CI-CPS models, the generated code relies on Simulink library (*.lib) files and the symbolic executor cannot handle those external libraries. That is the main reason SLDV performs worse than SPsCGF. In other small or medium control logic models, SPsCGF performs slightly better than SLDV. That is because SPsCGF and SLDV both use symbolic execution to solve constraints. In small or medium models, the symbolic execution is still the state-of-art method to generate test cases. Even in small or medium models, if the number of loops in the model is high, SLDV cannot generate test cases efficiently. For model *Regulator* and model *NLGuidance*, the number of loops in those model is more than 100, SLDV takes much more time (46s-170s) than expected to generate test cases. If the code in a loop should be executed more than 1000 times, SLDV often takes more than 20 min to generate test cases. In the meanwhile, the bounded model checking (BMC) method can generate test cases in seconds because it only expands the loop for a finite number of times. That is why SPsCGF-BMC, which is the bounded model checking technique used in SPsCGF, achieves more than 17% better coverage than SLDV.

**Table 1.** Effectiveness.

| | Condition& Decision Coverage | | | |
|---|---|---|---|---|
| **Model** | **SPsCGF-BMC** | **SPsCGF** | **SLDV** | **SimCoTest** |
| NLGuidance | 38% | 69% | 38% | 31% |
| RHB1 | 0% | 89% | 0% | 81% |
| RHB2 | 0% | 91% | 0% | 85% |
| Euler321 | 94% | 94% | 50% | 47% |
| BasicTwoTanks | 100% | 100% | 96% | 53% |
| EB | 98% | 98% | 93% | 0% |
| MHI1209 | 0% | 96% | 0% | 92% |
| AFC | 0% | 67% | 0% | 67% |
| AT | 0% | 79% | 0% | 68% |
| IGC | 0% | 100% | 0% | 96% |
| Regulator | 75% | 75% | 64% | 50% |
| BIMultiplexor | 88% | 88% | 88% | 69% |

For small or medium control logic models, SPsCGF performs significantly better than SimCoTest because SPsCGF uses symbolic execution to solve constraints while SimCoTest solely uses the random testing. For complex non-CI-CPS models, the symbolic execution technology has a poor performance. The coverage achieved by SPsCGF is higher than that of SimCoTest, about 4% to 38% in non-CI-CPS models. The main reason for the coverage improvement is that SPsCGF runs faster than SimCoTest. The speed of SimCoTest is slow, the SPsCGF fuzzing method can generate tens of thousands of test cases in a second, which is the reason SPsCGF fuzzing method performs better than SimCoTest. The SimCoTest does not rely on parallel execution while SPsCGF does; given the same testing time, SPsCGF performs averagely 25% better than SimCoTest without considering the extreme cases. In the future, the taint analysis technology in fuzzing can be turned on to further improve the testing efficiency in model testing. Because SimCoTest relies on Simulink simulation, in practice, the running overhead of SimCoTest is much larger than SPsCGF which is based on source code. The comparison about the speed of each tool will be described in next subsection. In overall results, without considering the most extreme cases, SPsCGF performs 8% to 38% better than SimCoTest and performs 11% to 31% better than SLDV.

### 4.2. RQ2—Efficiency

The execution times of SPsCGF, SLDV and SimCoTest for the subject models in the benchmark are shown in Table 2. To be fair, we collect the time taken for each tool to reach a certain coverage. This certain coverage is the lowest coverage achieved by the three tools.

The results show that SPsCGF is significantly more efficient than SLDV or SimCoTest. The SLDV is much slower than expected, it takes too many times to do symbolic analysis. If the code in a loop should be executed more than 1000 times, SLDV often takes more than 20 min to generate test cases. In many cases, SLDV is too slow to generate test cases within the given time. Loops in a model have a significant impact on the result of the symbolic execution, so we use the bounded model checking to solve this problem to improve the testing speed. This explains why SPsCGF is much faster than SLDV.

The SimCoTest is also slower than SPsCGF. In most situations, SimCoTest does not stop at the preset time. In most cases, SPsCGF only takes 10–20 s to accomplish a task while SimCoTest takes 2–3 min to accomplish the same task. As the state-of-the-art tools such as SimCoTest depend on the simulation environment of Simulink, there exists a running overhead for Simulink simulation, which decreases the efficiency of testing. In contrast, the SPsCGF relies on generated code to do fuzzing testing and symbolic execution, this contributes to the high speed of testing. Furthermore, SPsCGF follows the fuzzing framework which supports parallel execution of the model, while SimCoTest do not support parallel execution. This is becoming less and less competitive when multi-core computers become more and more popular. From the results, we conclude that the fuzzing method has great advantages in efficiency. In the future, the code optimization techniques can be used to continue improving testing efficiency.

**Table 2.** Time Efficiency.

| | Time to Achieve Same Coverage | | |
|---|---|---|---|
| **Model** | **SPsCGF** | **SLDV** | **SimCoTest** |
| NLGuidance | 10 s | 171 s | 118 s |
| RHB1 | 5 s | NA | >180 s |
| RHB2 | 4 s | NA | >180 s |
| Euler321 | 2 s | 46 s | 152 s |
| BasicTwoTanks | 11s | >180 s | 143 s |
| EB | 15 s | 79 s | NA |
| MHI1209 | 27 s | NA | 127 s |
| AFC | 17 s | NA | >180 s |
| AT | 52 s | NA | >180 s |
| IGC | 7 s | NA | 120s |
| Regulator | 12 s | >180 s | 124 s |
| BIMultiplexor | 2 s | 12 s | 96 s |

*4.3. RQ3—Usefulness*

We use the MHI1209 model for the industrial case study. This model takes 8 input ports and 54 parameters. A small part of the model is shown in Figure 7. Note that the Figure 7 is only a small fragment (1/50) of the original large model, some parts may not be displayed completely. The *COMP* and *ONDLC* are special Simulink components containing the user-written code. In this experiment, we compare the coverage results by using two methods: SPsCGF method and the raw coverage guided fuzzing (rawCGF) method. The only difference between these two methods is that SPsCGF uses mutation operators based on signal patterns. The experiment is conducted on a machine with 32G memory, i5 6-cores, 2.8GHz CPU, and Win10 OS.

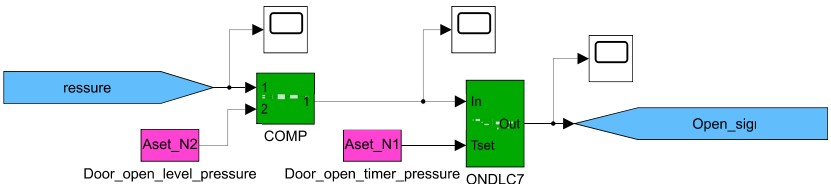

**Figure 7.** Use Case.

The MHI1209 model contains elements which SLDV cannot handle, thus the comparison of SLDV is omitted here. Figure 8 shows the coverage results over time. As can be seen from Figure 8, both tools will improve coverage in 50 s. It is difficult for two tools to achieve a significant coverage increase after 60 s. From the figure, compared to the rawCGF method, we can see that SPsCGF can improve the coverage and the mutation operators based on signal patterns take effect. For SPsCGF method, because we use more mutation operators compared to the rawCGF method in fuzzing, there is a higher possibility that the SPsCGF method can achieve higher coverage, which is proved in the Figure 8. In Figure 8, around 40 s, the rawCGF method stopped improving coverage, but SPsCGF method can continue to achieve higher coverage. According to the Figure 8, the improvement we have achieved seems to be very small, but that is because the entire model is particularly large. Those components that are very easy to be covered account for the most of the entire model. If we exclude those components that are particularly easy to be covered, there will be a significant difference in the comparison of coverage rates.

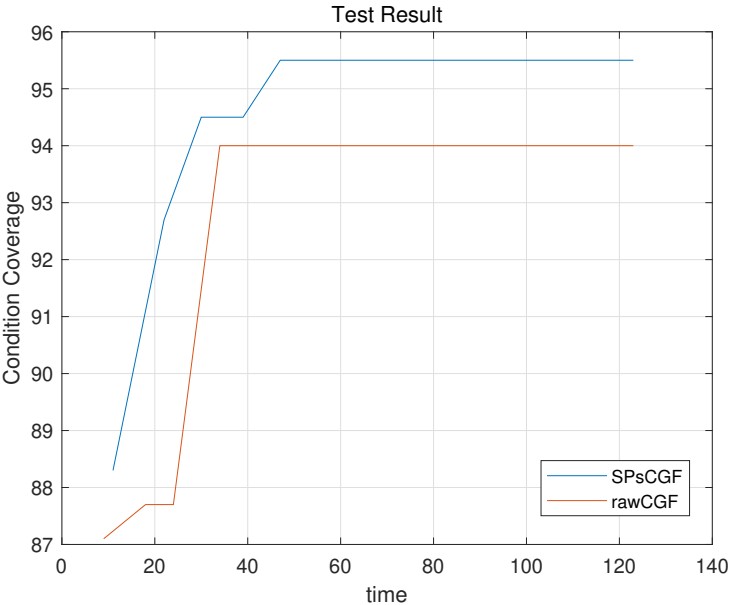

**Figure 8.** Coverage Diagram in MHI Model.

Thus, we further examine the elements covered by SPsCGF but not covered by rawCGF. Those elements are shown in Table 3. The logic of *ONDLC* has been shown in Figure 1b and *OSHOTC* has a similar logic pattern as *ONDLC*. As can be seen, in those elements, the coverage difference is huge. This confirms that the mutation operators based on signal patterns take effect in fuzzing procedure.

**Table 3.** Case Study.

| | Differences in Details | |
| :---: | :---: | :---: |
| **Component** | **SPsCGF** | **SimCoTest** |
| OSHOTC1 | 86% | 100% |
| OSHOTC13 | 86% | 100% |
| OSHOTC14 | 86% | 100% |
| ONDLC3 | 40% | 100% |
| ONDLC6 | 40% | 100% |

For *Modified Condition/Decision Coverage* (MC/DC) coverage, the decision expressions in most models only have one condition. In this case, the MC/DC coverage is equivalent to condition coverage or decision coverage. In benchmarks, there are only two models

that contain two or more decisions which consists of two or more conditions. The only two models are NLGuidance and Euler321. The MC/DC coverage is collected on these models. As shown in Table 4. SPsCGF-BMC performs well because CBMC (the kernel of SPsCGF-BMC) has an efficient built-in algorithm specially designed for MC/DC coverage. Even so, the fuzzing method achieves higher MC/DC coverage based on the initial seeds generated by CBMC.

**Table 4.** MC/DC Coverage.

| | MC/DC Coverage | | |
|---|---|---|---|
| **Model** | **SPsCGF-BMC** | **SPsCGF** | **SLDV** |
| NLGuidance | 0% | 20% | 0% |
| Euler321 | 57% | 76% | 0% |

## 5. Related Work

Formal verification techniques [5–9] aim to exhaustively check the correctness of models, but they often face scalability issues for complex CPS models. Simulink Design Verifier [10] is the representative tool which uses formal verification techniques. For small and medium models, Simulink Design Verifier can achieve high coverage, but for large models, Simulink may fail to generate test cases. In order to verify large systems, The counterexample-guided abstraction refinement (CEGAR) frameworks [29–34] have been proposed. Those techniques abstract hybrid system models into discrete finite state machines without dynamics or replace the complex system with a simpler one. However, these techniques need human intervention as the abstraction of the models need to be predefined. Search based techniques [12–16,18,19,22] such as genetic algorithms or guided simulation algorithm [17] are also widely used in model testing to obtain high coverage. The concolic testing [35] is also applied to Simulink model to ensure the model safety. To generate inputs to find defects or wrong output signal patterns, different genetic algorithms [22,36] have been proposed. Those algorithms use different fitness functions and fitness definitions to guide the search. The search-based techniques can also be used to approximate the system [37,38] to help generate test cases. To evaluate the effectiveness of the test cases, mutation testing techniques [39,40] have also been applied to models. By injecting manually created faults into models, the effectiveness of test cases can be evaluated according to whether the test cases can find the manually created faults.

## 6. Limitation and Future Work

When dealing with models using current testing methods, if the model relies on external binary libraries, the final testing effect is poor because the existing symbolic execution tools cannot track the code in binary format, thus, the constraints in external binary libraries cannot be collected and solved. Actually, existing model-checking tools cannot process binary files at all, while the dynamic symbolic execution tool KLEE can process binary files, but it needs to decompile the binary files into the IR of LLVM, which in many cases is particularly complex and cannot be automated. Moreover, KLEE cannot support Windows operating systems, and many projects require tools to support Windows operating systems. In future work, we need to extend our bounded model checking and fuzzy testing methods to binary testing in order to better support industrial scenarios. Although we have implemented a simple binary instrumentation and fuzzy testing framework that can run, The binary level analysis is particularly difficult. A binary executable file contains a large number of system library instructions, most of which we do not need to handle. Handling invalid instructions can result in slow testing speeds. How to improve the testing efficiency of binary files is also an important research direction for the future.

## 7. Conclusions

We presented a framework that combines bounded model checking with coverage-guided fuzzing in a novel way to generate test cases. In the given benchmarks, the proposed framework can achieve 8% to 38% coverage improvement and $3\times$–$10\times$ speed improvement compared with the state-of-the-art baselines. Existing works have serious efficiency problems in industrial scenarios, and the proposed method can effectively improve testing efficiency. For the current tool implementation, users are required to provide source files, such as source code or model source design files. However, in many industry cases, some critical models or systems are in binary format. Because of copyright, no source code or model source file is provided. How to directly test binary code is an urgent problem to be solved in the future. In order to target the actual harsh industrial scenarios, in the next study, we will continue to delve into binary-level model testing.

**Funding:** This research was funded by National Natural Science Foundation of China (NSFC) grant number 62102220 and the APC was funded by [NSFC-Youth 62102220].

**Institutional Review Board Statement:** Not applicable.

**Informed Consent Statement:** Not applicable.

**Data Availability Statement:** The code and the data set are available at https://github.com/EmbedSystemTest/SimulinkTest.

**Conflicts of Interest:** The authors declare no conflict of interest.

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
