# Peer review of "Improve Model Testing by Integrating Bounded Model Checking and Coverage Guided Fuzzing"

_electronics, doi:10.3390/electronics12071573_

Round 1

Reviewer 1 Report

The article is quite detailed and describes the subtle aspects of the testing problem. The author has covered all aspects of the new testing system SPsCGF and its implementation.

In Fig. 6 there is a typo "ressure", also re-proofreading of text is required.

To improve the article, it may be useful to add more results such as Fig.7.

Also, it is useful to specify the type of processor and operating system on which Fig.7 was obtained.

Author Response

  1. I have added the type of processor and operating system for original Fig. 7. 
  2. I have added more discussions in experiments. 
  3. For original Fig. 6, it is a problem of MATLAB Simulink that makes the long text in a widget not display completely. For each long text in a widget in UI of Simulink, the display effect is like Fig. 6. Besides, the original Fig. 6 is just a small screenshot of the original large model, I have added the explanination around the figure to explain that this is only a small fragment (1/50) of the original large model, some parts may not be displayed completely. Thanks for pointing this. 

Reviewer 2 Report

The paper demonstrates the effectiveness of the proposed approach in improving model testing and achieving higher model coverage and time efficiency compared to existing methods. It seems like a valuable contribution to model testing for electromechanical systems to address the challenges in ensuring safety and security in complex industrial cases.

There are some significant grammatical mistakes in the paper which need to be addressed. I would suggest to do that before resubmitting.

Author Response

I have slightly reorganzied the content of the paper, delete some unnecessary content, add some details and change some mistakes as far as I can.  If the paper still needs improvement, please inform me and I will use MDPI English Editing Service to improve my paper. 

Reviewer 3 Report

Recommendation: Major Revision 

1.        The article could benefit from a clearer introduction. It is not immediately clear what the main purpose of the article is and what the problem is that the authors are trying to solve. It would be helpful to provide a clear and concise statement of the problem and the solution that the authors propose.

2.        The introduction provides good background information, but it is too lengthy and lacks a clear research question or objective.

3.        The authors should provide a more thorough explanation of the proposed SPsCGF framework, including the mutation operators used in the fuzzing stage.

4.        The results section should provide more detailed information on the benchmarks used and the specific improvements achieved by the proposed method compared to other methods.

5.        The authors could discuss potential limitations or challenges of the proposed method and future directions for research.

Author Response

  1. I have reorganized the Intriduction, the first paragrah shows the background, the second paragraph shows the first research problem, the third paragram shows the second research problem, the fourth paragraph shows the proposed solution to these problems mentioned in previous., the fifth paragraph shows the details of the proposed framework. I hope the new organization can ease the reading. 
  2. I have deleted some contents in Introduction and slightly added the objective of the proposed method. 
  3. I have added the overall work flow Figure and necessary description about how the mutation works in subsection: Coverage Guided Fuzzing. The newly added content make up nearly one page, each mutator description has been appended with a detailed example. 
  4. I have added the more description about the benchmark in The models in benchmark and some more details about the experiments. 
  5. I have added a subsection about the limitation and the future direction. 

Round 2

Reviewer 3 Report

Recommendation: Accept